# Learning Generalised Policies for Numeric Planning

**Primary Keywords:** *(2) Learning;*

## Abstract

We extend Action Schema Networks (ASNets) to learn generalised policies for numeric planning, which features quantitative numeric state variables, preconditions and effects. We propose a neural network architecture that can reason about the numeric variables both directly and in context of other variables. We also develop a dynamic exploration algorithm for more efficient training, by better balancing the exploration versus learning tradeoff to account for the greater computational demand of numeric teacher planners. Experimentally, we find that the learned generalised policies are capable of outperforming traditional numeric planners on some domains, and the dynamic exploration algorithm to be on average much faster at learning effective generalised policies than the original ASNets training algorithm.

## Introduction

*Generalised planning* is broadly concerned with the representation, synthesis, and learning of plans, policies, heuristics, and other forms of control knowledge applicable to *many* problem instances (Srivastava, Immerman, and Zilberstein 2011; Hu and Giacomo 2011; Celorrio, Aguas, and Jonsson 2019). Interest in generalised planning has steadily increased in recent years, fueled in part by advances in machine learning, and by the development of new formalisms to represent and reason about generalised planning tasks and their solutions (Toyer et al. 2018; Francès et al. 2019; Garg, Bajpai, and Mausam 2019; Bonet and Geffner 2020; Aguas, Jiménez, and Jonsson 2020; Toyer et al. 2020; Shen, Trevizan, and Thiébaux 2020; Karia and Srivastava 2021; Aguas, Jiménez, and Jonsson 2021; Ståhlberg, Bonet, and Geffner 2022; Lin et al. 2022).

An important limit of existing work on generalised planning is that it only allows for primitive forms of quantitative information to be modelled,[1] even though such information is core to many real world problems – for example, modelling a delivery robot requires modelling how much weight it can hold, and modelling flights require reasoning about the the product of distance travelled and fuel consumption per unit of distance. This is despite the existence of the vibrant field of *Numeric planning*, which extends classical planning formalisms to allow modelling numeric fluents, conditions and effects (Fox and Long 2003), and typically handles

---

[1]We are only aware of a single exception (Lin et al. 2022).

them using new heuristic search, optimisation, or satisfiability modulo theory based techniques (Hoffmann 2003; Coles et al. 2013; Scala et al. 2016a,b, 2020; Kuroiwa et al. 2022; Leofante 2023).

In this paper, we extend a state of the art generalised planning approach, namely *Action Schema Networks* (ASNets) (Toyer et al. 2018, 2020) to handle numeric planning problems described in PDDL2.1 (Fox and Long 2003). ASNets is a recent deep learning architecture capable of representing policies for generalised planning, and designed to learn from smaller planning tasks and then apply that knowledge to tackle larger, more complex challenges within the same domain. The network's contruction exploits the relational structure of planning problems and domains, and its connectivity reflects the precondition-effect relationships captured in the domain's action schemas. This scheme makes it possible to share weights between policy networks instantiated for different problems in a domain, and learn a single set of parameters which can be transferred to problems of arbitrary size in that domain. ASNets are trained using an imitation learning algorithm, which iteratively explores the state space of the training problems using a teacher planner. Experiments with classical and probabilistic domains have shown that ASNets can outperform conventional planners when the domain has simple tricks that are key to solving larger problems but which can be learned from small problems.

To extend ASNets to numeric planning, We first propose a network module that enables ASNets to directly reason about numeric fluents. Then, we illustrate why such reasoning is not always sufficient for learning effective generalised policies. We argue that reasoning about the interaction between individual fluents is crucial, and allow ASNets to perform this interaction reasoning through numeric comparisons in the problem. To cope with the increased length of numeric plans and run-time of numeric planning teachers in comparison with their classical planning counterparts, we also propose a new training algorithm which offers greater control over the exploration versus learning balance.

Finally, we evaluate our proposed techniques on a representative set of benchmarks from the latest International Planning Competition, featuring both simple and linear numeric planning domains. Our results show that our extensions to ASNets allow it to learn generalised policies capa-

ble of solving problem instances significantly more complex than those seen in training, and outperform non-learning planners in several domains. We also find that the greater control offered by our new training algorithm allows ASNets to be trained much more quickly without compromising the effectiveness of the learned generalised policies.

## Numeric Planning

As in PDDL2.1 (Fox and Long 2003), a numeric planning problem, denoted as $P = \langle D, I \rangle$, consists of a *domain* $D$ and an *instance* $I$. The domain includes predicates $\mathcal{P}$, functions $\mathcal{F}$, and action schemas $\mathcal{A}$; the instance $I$ comprises objects $O$ and additional elements. Each predicate $p \in \mathcal{P}$ and function $f \in \mathcal{F}$ applies to object arguments $o_1, \ldots, o_n$ from $O$ to form ground proposition $p(o_1, \ldots, o_n)$ and fluents $f(o_1, \ldots, o_n)$ respectively.[2] Through grounding, the sets $\mathcal{P}, \mathcal{F}$, and $O$ define the set of all possible propositions $P$ and fluents $F$, which encode a state space $S$ where a state is an assignment of boolean values to each proposition and real values to each fluent.

A *comparison schema* has the form $\xi \trianglerighteq \gamma$, where $\trianglerighteq \in \{\leq , <, =, >, \geq\}$, $\xi$ is an arithmetic expression over $\mathcal{F}$, and $\gamma$ is a real constant. Once grounded, $\xi$ becomes an arithmetic expression over $F$, and the ground *comparison* is a mapping from $S$ to a truth value. Each action schema $\alpha \in \mathcal{A}$ has a precondition $\mathrm{pre}(\alpha)$ that is a conjunction of comparison schemas and predicates. The effect of $\alpha$, $\mathrm{eff}(\alpha)$, is a schema to assign boolean values to propositions and/or increase/decrease/assign the value of arithmetic expressions over $\mathcal{F}$. Given objects $O$, action schemas in $\mathcal{A}$ ground to a set of actions $A$. The problem $P$ is *linear* if all arithmetic expressions are linear, and *simple* if furthermore the numeric action effects only involve increasing or decreasing fluents by a constant.

The instance $I$, in full, is a tuple $I = \langle O, s_0, G, M \rangle$. The initial state $s_0$ is any state in $S$, the goal $G$ is a conjunction of comparisons and propositions, and the plan metric $M$ is an optional arithmetic expression over $F$. An action is applicable in a state when its precondition is satisfied, and its application yields a new state according to its effect. Propositions and fluents not included in the effect remain unchanged. A plan is a sequence of actions. It is an *executable* plan if when iteratively applied in $s_0$, each subsequent state satisfies the next action precondition, and it is a *goal achieving* plan if the final state satisfies $G$. A *valid* plan is an executable goal achieving plan. The cost of a valid plan is the value of $M$ at the final state if $M$ is provided, or otherwise the number of actions in the plan. For this paper, a *generalised policy* for a domain $D$ is a mapping from instances to executable (but not necessarily goal achieving) plans. The *effectiveness* of a generalised policy over a finite set of instances measures the proportion of instances the policy maps to goal-acheving plans. The more effective a generalised policy is, the larger this number is.

---

[2]The value of $n$, or *arity*, is dependent on the particular predicate or function.

## Numeric Action Schema Networks

Action Schema Networks are a state of the art approach for learning generalised policies for classical planning problems (Toyer et al. 2020). Core to its effectiveness is its ability to generalise from a small set of training problems to much larger and unseen problems in the same domain, thereby amortising training time. This original approach is unable to perform numeric reasoning effectively due to a lack of architectural components dedicated to numeric reasoning. In this section we propose *Numeric Action Schema Networks* ($\nu$-ASNets) for learning generalised policies for numeric planning.

A $\nu$-ASNet is a neural network with weights $\theta$ that takes in input vectors describing the current state $s$ and outputs a probability distribution $\pi^\theta(a|s)$ over all applicable actions $a$. For each problem instance, a $\nu$-ASNet is constructed with the weights $\theta$ shared between all instances in the same domain. That is, the network architecture is *instance-dependent*, but the weights are *instance-agnostic* through a *weight-sharing* mechanism that we describe later.

For each instance, the $\nu$-ASNet architecture includes layers of network modules that alternate between encoding action and state information, as shown in Figure 1. Each *action layer* contains an *action module* for each action $a \in A$. The last layer of a $\nu$-ASNet is always an action layer whose outputs determine $\pi^\theta$, and we inherit the assumption from ASNets that the first layer is also always an action layer. Each *state layer* contains one *state module* for each piece of state information, namely propositions, fluents, and comparisons. Each network fixes a hidden dimension $d_h$, and networks modules propagate forward a hidden representation vector in $\mathbb{R}^{d_h}$ to connected modules in the next layer. Network modules are connected sparsely to modules in adjacent layers through a notion of relatedness.

**Definition 1 (relatedness)** *An action $a$ is* related *to a proposition $p$, fluent $f$, or comparison $c$ at position $k$, denoted by $R(a, p/f/c, k)$, if $p/f/c$ is a ground instance of the $k$th unique predicate/function/comparison schema appearing in the action schema of $a$, respectively.*

**Example 1** *Consider the following action schema for a robotic arm picking up an object of a given weight, subject to a limit on the total load carried by the arm:*
$\mathrm{pickup}(b, o)$*:*
$\mathrm{prec} : \mathrm{clear}(o), \mathrm{weight}(o) + \mathrm{load}(b) \leq \mathrm{limit}(b)$
$\mathrm{eff} : \neg\mathrm{clear}(o), \mathrm{holding}(b, o), \mathrm{load}(b) \mathrel{+}= \mathrm{weight}(o)$
*The action* $\mathrm{pickup}(b_1, o_1)$ *for a particular robot arm* $b_1$ *and object* $o_1$ *is related at position 1 to the proposition* $\mathrm{clear}(o_1)$, *at position 2 to the proposition* $\mathrm{holding}(b_1, o_1)$, *at position 1 to the fluent* $\mathrm{weight}(o_1)$, *at position 2 to the fluent* $\mathrm{load}(b_1)$, *at position 3 to the fluent* $\mathrm{limit}(b_1)$, *and at position 1 to the comparison* $\mathrm{weight}(o_1) + \mathrm{load}(b_1) \leq \mathrm{limit}(b_1)$.

The notion of relatedness extends naturally to the various network modules introduced below.

**Action modules.** The action module for $a \in A$ in the $l$th action layer takes an input vector $u_a^l$ and produces a hidden representation

$$\phi_a^l = \sigma(W_a^l \cdot u_a^l + b_a^l)$$

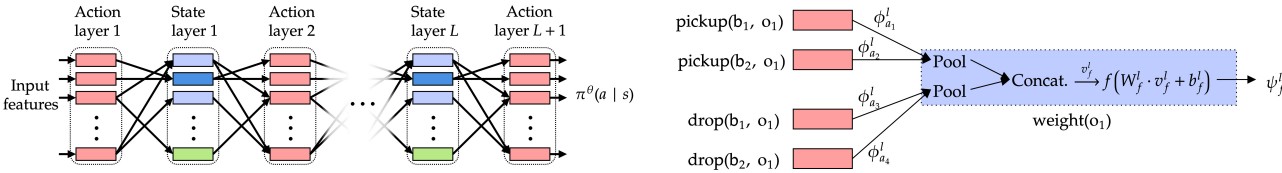

Figure 1: (Left) Overview of an $\nu$-ASNet with $L$ state layers and $L+1$ action layers, with colours in state layer indicating the different types of modules. (Right) example fluent module for the fluent $\text{weight}(o_1)$ in a problem instance with two robot arms $b_1$ and $b_2$ from a domain where $\text{weight}(o)$ occurs in two action schemas $\text{pickup}(b, o)$ and $\text{drop}(b, o)$.

where $W_a^l \in \mathbb{R}^{d_h \times d_a^l}$ and $b_a^l \in \mathbb{R}^{d_h}$ are the learnt weight matrix and bias vector for the action module, $\sigma$ is a non-linearity, $d_h$ is the fixed hidden representation size, and $d_a^l$ is the dimension of $u_a^l$. The input vector is constructed by concatenating the hidden representation of all related state modules in the previous layer, ignoring relatedness position

$$u_a^l = \left[\psi_1^{l-1}{}^T \ldots \psi_M^{l-1}{}^T\right]^T$$

where $\psi_j^{l-1}$ is the hidden representation produced by a related state module in the preceding state layer. Since all $\psi_j^{l-1}$ have dimension $d_h$, $u_a^l$ has dimension $M \cdot d_h$.

The related state modules of an action $a \in A$ can be determined by enumerating all the predicates, functions, and comparison schemas in its action schema grounding them using the same objects used to ground $a$. If we impose an ordering on these constructs (e.g. using position), the structure and dimension of $u_a^l$ across all actions with the same schema is fixed. Such structure is the key to weight-sharing. For a given domain, all action modules across different $\nu$-ASNets at the same layer $l$ with the same action schema $\alpha$ share the same weight matrix $W_\alpha^l$ and bias vector $b_\alpha^l$. This allows us to apply the same set of weights to any instance in a domain, as all actions in these problem instances are grounded from the same set of action schemas.

**Fluent modules.** In each state layer, there is one fluent module for each fluent in the problem. Fluent modules allow the network to reason directly about the quantitative components of the state space. Like action modules, a fluent module for fluent $f \in F$ in the $l$th state layer computes a hidden representation

$$\psi_f^l = \sigma(W_f^l \cdot v_f^l + b_f^l)$$

where $W_f^l \in \mathbb{R}^{d_h \times d_f^l}$ and $b_f^l \in \mathbb{R}^{d_h}$ are the learned weight matrix and bias vector, $\sigma$ is the same nonlinearity as before, $v_f^l$ is the input feature vector, and $d_f^l$ is the dimension of $v_f^l$.

Like action modules, weight-sharing requires that the input vectors $v_f^l$ have a similar structure for fluents derived from the same function. Unlike action modules, the number of actions related to a fluent is not instance-agnostic, so simple concatenation of hidden representations of related action modules from the preceding layer is not sufficient. We treat this similar to how proposition modules are constructed in the original ASNets. From all the actions related to $f$ at

various positions, we extract their action schemas and enumerate all unique pairs $\{(\alpha_1, k_1), \ldots, (\alpha_S, k_S)\}$ of action schemas and position pairs. These pairs are only dependent on the function of $f$. We then construct the input feature by

$$v_f^l = \begin{bmatrix} \text{pool}(\{\phi_a^l \mid \text{op}(a) = \alpha_1 \wedge R(a, f, k_1)\}) \\ \vdots \\ \text{pool}(\{\phi_a^l \mid \text{op}(a) = \alpha_S \wedge R(a, f, k_S)\}) \end{bmatrix}$$

where $\text{op}(a)$ is the action schema of $a$ and $\text{pool}$ is a pooling function to combine multiple $\mathbb{R}^{d_h}$ vectors into one $\mathbb{R}^{d_h}$ vector. Like the original ASNets we use the element-wise max function for $\text{pool}$. The structure of the resulting input feature is only dependent on domain information, namely the action schemas and functions, and hence enables weight-sharing – all fluent modules across different $\nu$-ASNets at the same layer $l$ with the same function share the same weight matrix and bias vector.

**Proposition modules.** Like the original ASNets, in each state layer we include a proposition module for each proposition in the problem. Proposition modules are almost identical to fluent modules, with the same computation for hidden representation, construction of input feature, and weight-sharing property.

**Comparison modules.** By including fluent modules in the network and fluent values in the network input, $\nu$-ASNets are able to learn generalised policies that reason directly on the value of each fluent. Such reasoning is unfortunately not always sufficient. Consider a domain $D$ whose sole numeric component involves robotic arms with load limits lifting up items of varying weight, and suppose two problem instances $I$ and $I'$ differ only in that all the load limit and item weights in $I'$ are double their counterparts in $I$. There is no practical difference between the problem $P = \langle D, I \rangle$ and $P' = \langle D, I' \rangle$, and an ideal generalised policy should produce the same plan for both problems. Fluent modules are unable to recognise this "symmetry" between $P$ and $P'$ – weights learned by training on $P$ would not apply directly to $P'$.

More generally, the value of fluents are often only meaningful in the context of other fluent values, and it is valuable to allow learned generalised policies to reason about the interaction between fluents. In particular, fluents interact in comparisons in action preconditions, which we capture through *comparison modules*. Let $\text{comp}(a)$ denote the comparisons in action $a$ and $C = \bigcup_{a \in A} \text{comp}(a)$, in each state

layer there is one comparison module for each comparison. A comparison module for the comparison $c \in C$ in the $l$th state layer computes a hidden representation

$$\psi_c^l = \sigma(W_c^l \cdot v_c^l + b_c^l)$$

where $W_c^l \in \mathbb{R}^{d_h \times d_c^l}$ and $b_c^l \in \mathbb{R}^{d_h}$ are the learned weight matrix and bias vector, $\sigma$ is the same nonlinearity as before, $v_c^l$ is the input feature vector, and $d_c^l$ is the dimension of $v_c^l$.

Similar to fluent modules, a pooling mechanism is employed to enable weight-sharing between comparisons that share the same schema. From all actions related to $c$ at various positions, we extract their action schemas and enumerate all unique pairs $\{(\alpha_1, k_1), \ldots, (\alpha_S, k_S)\}$ of action schema and position pairs. Again, these pairs only depend on the comparison schema of $c$, and we construct the input feature using them by

$$v_c^l = \begin{bmatrix} \mathrm{pool}(\{\phi_a^l \mid \mathrm{op}(a) = \alpha_1 \wedge R(a, c, k_1)\}) \\ \vdots \\ \mathrm{pool}(\{\phi_a^l \mid \mathrm{op}(a) = \alpha_S \wedge R(a, c, k_S)\}) \end{bmatrix}$$

where $\mathrm{pool}$ is the same pooling function as before. The resulting input feature $v_c^l$ is again only dependent on domain information, and hence enables weight-sharing where all comparison modules across different $\nu$-ASNets at the same layer $l$ with the same comparison schema share the same weight matrix and bias vector.

**Input.** The first and last action layers take minor exceptions to the above as they are the input and output layers of the network. For the first layer, there is no preceding state layer and the input vector $u_a^1$ is a vector encoding state and heuristic information relevant to the current action. Specifically,

$$u_a^1 = \begin{bmatrix} v_p^T & v_f^T & v_c^T & g_p^T & g_f^T & m & c_a & c_{\mathcal{L}}^T \end{bmatrix}^T$$

where $v_p, v_f, v_c$ are the values of the related propositions, fluents, and comparisons of the action in the current state respectively, $g_p$ and $g_f$ indicate if the related propositions and fluents appear in the goal or not, $m$ is a boolean value indicating if the action $a$ is applicable in the current state, $c_a$ is the number of times $a$ has been applied so far, and $c_{\mathcal{L}}$ is a boolean vector encoding landmark information. For $v_c$, we treat the value of a comparison as the boolean value indicating if it is satisfied.

The lack of goal input $g_c$ for comparisons is a consequence of the lack of overlap between comparisons in action preconditions and comparisons in the goal. We also cannot include goal comparisons directly in the input as there is no notion of relatedness between them and actions. In domains where all instances of interest have goals with the same structure, one can define a "reach" action whose precondition is the original goal and effect is a proposition "goal-reached" which also replaces the goal. This special action would allow $\nu$-ASNets to reason about goal comparisons.

The inclusion of $c_a$ and $c_{\mathcal{L}}$ is to compensate for the *receptive field problem* discussed in the original paper (Toyer et al. 2020). Essentially, longest chain of related action and state modules the network can reason about is limited by its fixed

and finite depth. The inclusion of heuristic information can effectively address this problem. The action count $c_a$ helps the network avoid cycling between adjacent states. The numeric landmark encoding $c_{\mathcal{L}}$ in $\nu$-ASNets is derived from hybrid landmarks extracted from an AND/OR graph structure (Scala et al. 2017). Each such landmark $\ell$ has a target $t_\ell$ along with a set of actions $A_\ell$ and contributions for each action $\{\lambda_\ell^a \mid a \in A_\ell\}$, and represents the inequality

$$\sum_{a \in A_\ell} \lambda_\ell^a y_a \geq t_\ell$$

where $y_a$ is the number of times action $a$ is applied from the current state. Given a set of hybrid landmarks, the resulting landmark encoding $c_{\mathcal{L}}$ is a vector in $\{0, 1\}^3$, where $c_{\mathcal{L}}^{(1)} = 1$ if the action $a$ appears as the only action in $A_\ell$ for any landmark, $c_{\mathcal{L}}^{(2)} = 1$ if the action $a$ appears in any $A_\ell$ with other actions, and $c_{\mathcal{L}}^{(3)} = 1$ if $a$ does not appear in any $A_\ell$.

We have also experimented with other encodings of hybrid landmarks, specifically encodings that take into account the contribution and target of landmarks. We additionally experimented with removing all numeric components of the numeric problem and encoding the LM-cut landmarks of the resulting classical planning problem (Helmert and Domshlak 2009), which were used in the original ASNets. We did not find experimental success for either.

**Output.** The last layer of the network is the output layer. The output $\phi_a^{L+1}$ of each action module in the last layer is only a single real number, and the resulting output of the network is the masked softmax of all individual outputs,

$$\pi^\theta(a \mid s) = \frac{m_a \exp(\phi_a^{L+1})}{\sum_{a' \in A} m_{a'} \exp(\phi_{a'}^{L+1})}$$

where $m_a$ is a boolean mask of whether the action $a$ is applicable in the current state $s$, and $\pi^\theta(a \mid s)$ is the probability of selecting action $a$ in state $s$.

Given weights $\theta$ for a domain, the resulting generalised policy for a domain produces a plan for each instance by repeatedly selecting and applying actions using $\pi^\theta$, starting at the initial state, and terminating upon reaching a goal state, a state with no applicable action, or a fixed length limit. Like ASNets, we use $\pi^\theta$ during training by sampling from it, and during evaluation by greedily selecting the action with maximum probability and breaking ties deterministically.

**Miscellaneous.** We have introduced comparison modules and fluent modules together, along with their implications for network input and action modules. It is important to note that they can and are designed to be applied separately. We term the network with only comparison modules *C-ASNets* and the network with only fluent modules *F-ASNets*. This specialisation allows the network to focus on a particular form of reasoning, reduce computation burden, and potentially reduce overfitting. To disambiguate, we will use B-ASNets to refer to the network with both modules, and $\nu$-ASNets to refer to the collection of architectural variations.

It is worth noting that $\nu$-ASNets make no requirement on the form of comparisons or numeric effects appearing in the

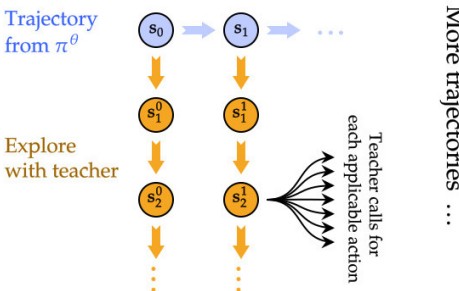

Figure 2: The exploration algorithm used by the original ASNets, where all states from the sampled trajectories are explored with the teacher planner. For all states shown, the teacher planner is called on all applicable actions to label the optimal actions

problem, and can be applied to the full numeric fragment of PDDL 2.1 (Fox and Long 2003). Additionally, we also include skip connections between modules that correspond to the same action, proposition, etc. (Toyer et al. 2020), which make it easier for the network to propagate information across layers.

## Dynamic Exploration

Like the original ASNets, for each domain, we train $\nu$-ASNets on a small number of training problem instances under the supervision and guidance of a teacher planner. We use the state-of-the-art ENHSP numeric planner as our teacher planner (Scala et al. 2020). The original algorithm trains over a number of epochs. Each epoch involves an *exploration phase* and a *learning phase*. The exploration algorithm used by the original ASNets first uses the current network weights $\theta$ to sample a number of trajectories from $\pi^\theta$, then explores all the states from these trajectories by calling the teacher planner on them, and adding all the resulting states to a multiset state memory $S_{\mathrm{mem}}$, as illustrated in Figure 2. For each state added to $S_{\mathrm{mem}}$, the teacher planner is called on the resulting state of applying each applicable action, and the actions leading to the lowest-cost plans are labelled optimal. The learning phase then updates the weights through mini-batch gradient descent to choose the optimal actions.

In our preliminary experiments we find that the original exploration algorithm is inadequate for effectively learning generalised policies for numeric planning. We observe that for numeric planning, the plan lengths of even simple training problems tend to be longer than that of classical planning, and ENHSP to be much slower than teachers used by ASNets for classical planning. In each epoch, the number of states added to $S_{\mathrm{mem}}$ in the original exploration algorithm is quadratic in the plan lengths of the training problems, resulting in a large number of states added for numeric planning. This has a number of downstream consequences in the original exploration algorithm:

1. For each state $s$ added to $S_{\mathrm{mem}}$, the teacher is called for each applicable action in $s$. An increased number of states added to $S_{\mathrm{mem}}$ therefore leads to a significant increase in calls to the teacher planner, especially in domains where states tend to have many applicable actions.

2. The learning phase uses states in $S_{\mathrm{mem}}$ for training. We observe that the size of $S_{\mathrm{mem}}$ is sometimes a few orders of magnitude larger than the number of states used in the learning phase. In this case many states added to $S_{\mathrm{mem}}$ are rarely used, wasting memory and exploration effort.

3. The alternating exploration and learning phases mean that states added to $S_{\mathrm{mem}}$ early are explored using outdated network weights (*stale*) and hence less useful for learning than more recently added states.

We propose a *dynamic exploration* algorithm to address these problems by removing stale states and dynamically adjusting the amount of exploration performed based on the time spent on the learning phase, as shown in Algorithm 1. In each exploration phase, dynamic exploration first uses the current network weights to generate $T_{\mathrm{gen}}$ trajectories for each training problem, i.e. calling the network iteratively starting at the initial state and sampling the action to apply from $\pi^\theta$ (line 8). Each such trajectory terminates upon reaching a goal, a fixed length limit, or a state with no applicable actions. The states in these trajectories are added to an initially empty $S_{\mathrm{traj}}$ multiset (line 3). We then repeatedly randomly remove (or *explore*) states from $S_{\mathrm{traj}}$ and call the teacher planner for each removed state until a termination condition is met (lines 12 to 13). All states in the resulting teacher plan are added to $S_{\mathrm{mem}}$ (line 14). This can be understood as asking the teacher planner to guide the network back onto a valid trajectory. If $S_{\mathrm{traj}}$ ever becomes empty, it is refilled by generating one trajectory from each training problem from the initial state.

The termination condition (line 9) is based on the average time $t_{\mathrm{learn}}$ spent on recent learning phases and an hyperparameter $r$ to control the ratio between spent on exploration and learning. We terminate exploration when either $r \cdot t_{\mathrm{learn}}$ time has elapsed in the current exploration phase or the number of states explored reaches an upper bound $e_{\mathrm{max}}$, but never before at least $e_{\mathrm{min}}$ states have been explored. For the first exploration phase where $t_{\mathrm{learn}}$ is undefined, we terminate once at lease one state from each problem and at least $e_{\mathrm{min}}$ states overall have been explored.

To avoid a size explosion of $S_{\mathrm{mem}}$ and ensure its states are recent, we group states in $S_{\mathrm{mem}}$ by the epoch they are added in. Whenever the number of states in $S_{\mathrm{mem}}$ exceeds a limit $N_{\mathrm{mem}}$, we repeatedly remove the oldest group till the size of $S_{\mathrm{mem}}$ falls back under the limit (line 16).

To address point 1, we also enable an option in the original ASNets implementation to approximate action optimality. Whenever a state $s$ is added to $S_{\mathrm{mem}}$, this option approximates the optimal action by only calling the teacher planner on $s$ instead of calling the teacher planning for each applicable action. The approximated optimal action is simply the action selected by the teacher planner on $s$.

## Experimental Evaluation

For evaluation, we implemented $\nu$-ASNets and dynamic exploration based on the original implementation of ASNets.

**Algorithm 1:** Dynamic exploration. We group new states in $S_{\mathrm{expl}}$ and add to $S_{\mathrm{mem}}$ with epoch number

---

**Data:** A set of training problems $P_{\mathrm{train}}$; current $\nu$-ASNets weights $\theta$; epoch number $n$

**1 Procedure** genTraj()
**2**    **for** $\zeta \in P_{\mathrm{train}}$ **do**
**3**      $S_{\mathrm{traj}}$.extend(runPolicy($s_0(\zeta), \pi_\theta$))

**4 Procedure** explore($n, S_{\mathrm{mem}}$)
**5**    $S_{\mathrm{traj}} \leftarrow \emptyset$
**6**    $S_{\mathrm{expl}} \leftarrow \emptyset$
**7**    **for** $i = 1, \ldots, T_{\mathrm{gen}}$ **do**
**8**      genTraj()
**9**    **while not** terminate() **do**
**10**      **if** $|S_{\mathrm{traj}}| = 0$ **then**
**11**        genTraj()
**12**      $s \leftarrow S_{\mathrm{traj}}$.popRandom()
**13**      $S_{\mathrm{expl}}$.extend(teacherPlan($s$))
**14**    $S_{\mathrm{mem}}$.extend($(S_{\mathrm{expl}}, n)$)
**15**    **while** $|S_{\mathrm{mem}}| > N_{\mathrm{mem}}$ **do**
**16**      $S_{\mathrm{mem}}$.popOldestEpoch()

---

The code will be publicly released when the paper is published.

## Experimental Setup

**Benchmark domains and teacher planner.** We use benchmarks from the International Planning Competition 2023 Numeric Track[3], and use the state-of-the-art numeric planner ENHSP-20 (Scala et al. 2020) as the teacher planner. These domains only include simple or linear numeric planning problems, and do not fully demonstrate the applicability of $\nu$-ASNetsto the entire numeric fragment of PDDL 2.1. For each domain, we use the 3 to 6 smallest instances for training. ENHSP has a wide set of configurations based on heuristic, search algorithm, and the use of methods such as redundant constraints and helpful actions. From these configurations, we select as the teacher ENHSP configuration one that produces short plans quickly (within one second) for the training instances. We do not experiment with domains where such a teacher configuration could not be found. The resulting benchmark domains and teacher configuration used for each domain are shown in Table 1. We classify domains by the proportion of reasoning that is numeric versus propositional into heavily numeric and hybrid domains.

For the domain Counters, we do not use the IPC instances, but instead use a set of evaluation instances with 2 to 60 counters where all the counters start with value 0. We only use one of these evaluation instances for training, and include for training another two instances similar to it but with different initial states. This set up allows us to better understand how the network would generalise across dimensions (number of counters) unvaried during training.

---

[3]https://ipc2023-numeric.github.io/

**Baselines and $\nu$-ASNet variations.** We use ENHSP as the baseline for comparison using all configurations that are used as teacher for at least one domain, and report its results for the best and teacher configurations of each domain. We also compare within the $\nu$-ASNets variations, namely the baseline network without either fluent or comparison modules (N-ASNets), F-ASNets, C-ASNets, and B-ASNets. We also compare the original ASNets training algorithm and the dynamic exploration algorithm for training F-ASNets and C-ASNets, and use superscripted $O$ or $D$ on the $\nu$-ASNets variation to denote them. For fairness, we enable action optimality approximation with the original algorithm.

**Hyperparameters.** For each domain, we train the network and evaluate it once on each problem instance. Unless otherwise specified, the hyperparameters for $\nu$-ASNets are fixed across domains and architectural variations. We use three action layers and two state layers, with a hidden representation size ($d_h$) of 15 and an ELU as the non-linearity $\sigma$ (Clevert, Unterthiner, and Hochreiter 2016). When using dynamic exploration, in each exploration phase we collect $T_{\mathrm{gen}} = 2$ trajectories initially, terminate exploration with parameters $r = 1$, $e_{\min} = 10$, and $e_{\max} = 1000$, and impose an $N_{\mathrm{mem}} = 15000$ limit on the size of $S_{\mathrm{mem}}$. When using the original algorithm, we collect two trajectories per problem and explore all states within the collected trajectories. After exploring, the learning phase performs weight optimisation using the Adam optimiser ($\beta_1 = 0.9$, $\beta_2 = 0.99$, and $\epsilon = 10^{-7}$). Mini-batch gradient descent is performed with a learning rate of 0.0003, batch size of 50, and 60 batches per epoch. We additionally apply an $\ell_2$ regulariser with a coefficient of 0.005 and a dropout probability of 0.1. We stop training when all collected trajectories reach the goal for 20 consecutive epochs.

**Computational limits.** When training, we apply a time limit of 8 hours for dynamic exploration and 24 hours for the original training algorithm. We apply a 1800 seconds time limit per problem for ENHSP and $\nu$-ASNets during evaluation. Training of $\nu$-ASNets and evaluation of ENHSP is performed on a virtual machine with 32GB of memory and a single dedicated core clocked at 4.5 GHz. Evaluation of $\nu$-ASNets is performed on the same virtual machine with only 8GB of memory.

## Results

Table 1 shows the coverage achieved by the $\nu$-ASNets variations and ENHSP by domain. The learned generalised policies are able to achieve coverages competitive with ENHSP, and outperform it on several domains, namely Delivery, FO-Counters, MPrime and TPP. Interestingly, except for FO-Counters, the other three domains all involve some forms of graph traversal and logistics. On Block Grouping, Rover, and Zenotravel, the $\nu$-ASNets achieve coverages commensurate with ENHSP. On the remaining two domains, Counters and Drone, $\nu$-ASNets are able to generalise from the small training problems to bigger problems and perform similarly with or outperform its teacher, but not the best ENHSP configuration.

| Domain | Classification | Teacher | Numeric ASNet | | | | | | | ENHSP | |
|---|---|---|---|---|---|---|---|---|---|---|---|
| | | | $B^D$ | $F^D$ | $C^D$ | $F^O$ | $C^O$ | $N^O$ | best | teacher | best |
| Block Grouping (20, 4) | HN, simple | hadd-gbfs | 15 (8.0) | 11 (8.0) | 17 (8.0) | 10 (15.5) | 15 (9.8) | 2 (24.0) | 17 | **20** | **20** |
| Counters (59, 1) | HN, simple | hrmax-astar | 9 (0.2) | 7 (0.1) | 14 (6.4) | 10 (0.1) | 17 (1.3) | 1 (14.8) | 17 | 8 | **39** |
| Delivery (20, 4) | hybrid, simple | hadd-astar | 5 (8.0) | 5 (5.3) | **20** (1.9) | 9 (9.9) | 18 (6.7) | 17 (3.3) | **20** | 8 | 16 |
| Drone (20, 4) | HN, linear | hadd-astar | 9 (8.0) | 4 (8.0) | 3 (8.0) | 7 (24.0) | 3 (24.0) | 0 (24.0) | 9 | 11 | **19** |
| FO-Counters (20, 3) | HN, linear | hrmax-astar | 4 (3.6) | 5 (1.6) | 3 (8.0) | **6** (7.3) | 3 (10.2) | 2 (15.4) | **6** | 4 | 5 |
| MPrime (20, 4) | hybrid, simple | hmrp-ha-gbfs | 16 (3.1) | **19** (1.8) | 12 (7.9) | 18 (4.4) | 16 (24.0) | 6 (24.0) | **19** | 16 | 18 |
| Rover (20, 4) | hybrid, simple | hmrp-ha-gbfs | **7** (8.0) | 4 (8.0) | 4 (8.0) | 5 (24.0) | 4 (24.0) | 4 (24.0) | **7** | **7** | **7** |
| TPP (20, 3) | hybrid, linear | hadd-gbfs | 0 (8.0) | 0 (8.0) | 19 (8.0) | 0 (24.0) | **20** (24.0) | 16 (22.7) | **20** | 4 | 4 |
| Zenotravel (20, 6) | hybrid, linear | hadd-gbfs | 0 (8.0) | 0 (8.0) | 17 (0.6) | 0 (24.0) | 16 (0.8) | 16 (0.6) | 17 | **20** | **20** |

Table 1: Number of instances solved (coverage) by each system, with the $\nu$-ASNets training time in hours shown in parenthesis. The number of instances for evaluation and the number of evaluation instances seen during training are shown in parenthesis after the domain. We also show the classification of the domain (see text) by heavily numeric (HN) versus hybrid (hybrid) and simple versus linear. We additionally show the teacher configuration used for each domain in the format {heuristic}−{search_algorithm}, with the optional "ha" indicating the use of helpful actions.

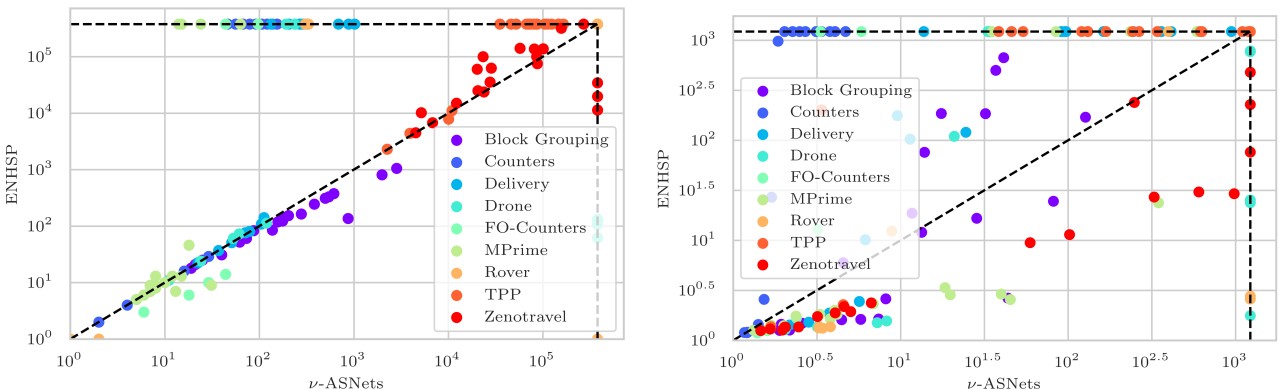

Figure 3: Plan cost (left) and runtime in seconds (right) for each problem instance of the best $\nu$-ASNets variation versus the teacher ENHSP configuration. Points in the bottom-right triangle favour ENHSP and on the top-left triangle favour $\nu$-ASNets. Problems unsolved by a system have value set to the maximum of the axis. A constant of 1 is added to ensure all points lie within view.

To better understand how well $\nu$-ASNets are able to generalise, we examine the particular problem instances to see if it is only generalising to problems with similar size to those seen during training. On Block Grouping, the largest training instance has 10 blocks on a 15 by 15 grid, whereas the largest solved instance has 25 blocks on a 100 by 100 grid. The training instances for Counters all have 4 counters, while $\nu$-ASNets variations are generally able to solve evaluation instances with up to 15 counters. This result on Counters shows that $\nu$-ASNets are able to generalise across factors (number of counters in this case) kept constant during training. The largest Delivery training instance has 10 items to deliver, while the largest solved instance has 42. Similar scales of generalisation are achieved on other domains, and demonstrate the strong generalisation capabilities of $\nu$-ASNets.

Table 1 also shows the training time of various $\nu$-ASNets variations. By comparing the training times of $C^D$-ASNets with $C^O$-ASNets and $F^D$-ASNets with $F^O$-ASNets, our re-

sults show that dynamic exploration is able to achieve much lower training times than the original exploration algorithm. This is not a consequence of the lower training time limit we apply for dynamic exploration, as the trend continues even when neither training methods reach their respective time limits, see e.g. in Delivery or FO-Counters. Furthermore, we do not observe any notable reduction in coverage for dynamic exploration when compared to training with the original algorithm. This suggests that dynamic exploration consistently enables learning generalised policies faster without compromising the effectiveness of the learned generalised policies. The only notable exception on Counters with C-ASNets is likely due to a high variance in training time that we found during multiple training runs.

Figure 3 shows the plan cost and evaluation runtime produced by the learned generalised policies and ENHSP with the teacher configuration. When both produce valid plans, they produce plans with similar costs. On Zenotravel $\nu$-ASNets tend to produce better plans, while on Block Group-

ing ENHSP tends to produce better plans. For runtime, when both produce plans quickly (less than 10 seconds), ENHSP tends to be quicker than the generalised policies. This is likely due to the higher constant overhead required by $\nu$-ASNets to construct the network and load the weights. On more complex problem instances, $\nu$-ASNets tend to produce plans faster than the ENHSP teacher configuration. The large number of points on the top line in the runtime plot demonstrates that the learned generalised policies are able to solve many instances the teacher cannot solve.

**Why do we need F-ASNets or C-ASNets?** Results in Table 1 show that the specialisation in reasoning offered by F-ASNets or C-ASNets often allow one of them to perform better than if they are both included. For example, comparison modules alone on Delivery or fluent modules alone in MPrime achieve notably higher coverages than when the other is included.

**Why do fluent modules result in coverages of 0 in TPP and Zenotravel?** In both domains, whenever fluent modules are included in the network, the learned generalised policies fail to solve any problem. In these domains, except for fluents used to help the plan metric, all the other fluents are only meaningful in context of each other. We suspect that when fluent modules are included, the network attempts to learn to reason directly on the fluent values, but receive conflicting information on how to do so on the different training instances. This results in training never being able to converge, and ultimately the coverage of 0.

**Which $\nu$-ASNets variation is the best?** The best $\nu$-ASNets variation by coverage depends on the nature of the domain. Generally, C-ASNets tend to perform well on all benchmark domains, while F-ASNets and B-ASNets perform well on particular domains such as MPrime and Rover respectively.

**How can N-ASNets perform well on some domains?** N-ASNets is not equipped with network modules for numeric reasoning, but it still has the capability for classical planning reasoning. Unsurprisingly, this allows it to still be effective on hybrid domains where there is a sizeable classical planning component. However, on heavily-numeric domains its unsuitability for numeric reasoning is clear from the poor coverage it achieves.

**How do $\nu$-ASNets compare with the IPC 23 competition planers?** On domains where we use the same problem sets as IPC 23 (i.e. all but counters), $\nu$-ASNets achieve better coverages than the reported coverage[4] of the IPC 23 competition planners except on Drone and Rover.

## Related Work

Existing work on generalised planning is severely limited when it comes to dealing with numeric information. Popular approaches based on Qualitative Numeric Planning (QNP) (Srivastava, Immerman, and Zilberstein 2011; Bonet and Geffner 2020), can represent a fixed number of positive

---

[4]https://ipc2023-numeric.github.io/results/presentation.pdf

numeric variables that can only be increased or decreased by a positive non-deterministic amount in action effects, and boolean combinations of comparisons of these variables with 0 in action preconditions, initial states and goals. Other approaches allow for incrementing or decrementing a finite set of positive counters by a constant in a deterministic fashion (Srivastava, Immerman, and Zilberstein 2010; Srivastava et al. 2015). The more recent Generalised Integer Numeric Planning (GLINP) (Lin et al. 2022) supports non-simple numeric effects, but is limited to integer variables. In contrast our work support the full numeric fragment (level 2) of PDDL2.1 (Fox and Long 2003), including nonlinear effects (Scala et al. 2016a) and numeric variables whose number grows with the number of objects. On the other hand, the above works provide guarantees on the effectiveness of generalised policies, whereas our learning approach cannot.

Independently and concurrently to our work, Tariq, Valenzano and Soutchanski (2023) experimented with handling numeric planning problems with the original ASNet policy representation. They reduced the set of values each fluent takes to a finite range, which they then manually discretised into consecutive intervals, each represented by a new predicate. Numeric conditions in the action schemas are then compiled into a disjunction over these predicates. As Tariq et. al observe, this approximation of the original numeric problem creates a large number of related propositions for each action, which leads to impactically large networks and compromises the sparseness of the ASNets policy representation. The empirical evaluation conducted by Tariq et al. only used four unseen test instances per domain. These are only marginally larger than the training instances and solved within less than a second by both ENHSP and ASNets. In contrast, we have proposed an architecture that treats fluents and comparisons as first-class citizens and explicitly reasons about them. Its performance is competitive with ENHSP over the latest numeric planning competition instances.

## Conclusion and Future Work

We have introduced $\nu$-ASNets and its variations, neural network architectures for learning generalised policies for numeric planning based on ASNets. The network is able to reason directly about numeric values through fluent modules, and about numeric contexts through comparison modules. We also introduced dynamic exploration, which trains $\nu$-ASNets much faster than the original algorithm used by ASNets, without harming the effectiveness of the learned generalised policies.

Our work leaves significant room for future research. A common trait in plans for numeric planning problems is the repetition of actions. We believe that a network architecture capable of not just predicting the action to apply, but also the number of times to apply it, can be highly effective. Additionally, while our work has focused on numeric planning, the method we use to construct comparison modules can in principle be applied to other components of actions modelled in PDDL, to include constructs such as action effects and universal/existential quantifiers. This leads to a network that can potentially learn generalised policies for a much more expressive class of problems than numeric planning.

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
