# OpenReview forum: "Learning Generalised Policies for Numeric Planning"
_icaps-conference.org/ICAPS/2024/Conference — ICAPS 2024_

### Official Review · Reviewer_cD32 · 2024-01-02

**Significance And Importance:** 2
**Soundness:** 3
**Novelty:** 3
**Clarity:** 3
**Overall Evaluation:** 2
**Confidence:** 3

**Weaknesses:**

2: No major or minor weaknesses.

**Contributions Of The Paper:**

The paper describes an extension of the ASNETs for tasks involving numeric state variables. Its main contributions lie in the definition of new modules in the architecture that can capture the numeric structure of the problem, made explicit by the presence of numeric constraints (comparison) and numeric effects. The ASNETs architecture is based on the idea of relatedness, a property that allows to link each action schema with the variables of the problem. Such a definition is extended to account for relatedness between fluents/numeric conditions and actions (I guess the term used in the paper for fluents is numeric fluents). This way, the network not only learns the potential influence of propositional variables, but also that given by particular values of numeric variables in a state. The network produces at the end a probability distributions over the applicable actions that can be used to generate a simple planner that greedily choose the most likely action until the goal, or to some fixed limit.

**Ethical Considerations:**

(1) Not Applicable: The paper does not have any ethical considerations to address

**Nomination For Best Paper:**

No

**Questions For Authors:**

1.The authors seem to assume that all preconditions and goals are expressed as conjunctions, but block-grouping is a domain which makes heavy use of disjunction. How is that accounted for?
2. How are ties broken when actions have the same probabilities?
3. Is it possible that the network produces a non zero probability for inapplicable actions? If not, can you please detail where this is enforced?

**Reproducibility:**

4: Authors promise to release code and domains (whichever apply).

**Strengths Of The Paper:**

1. A novel ASNETs architecture that is sensitive to the numeric structure of problems and generalises between different instances through a novel weighting passing mechanism
2. A thorough experimental analysis that compares different variants of the network
3. a novel training strategy that addresses the slowness of numeric planners vs classical planners.

**Weaknesses Of The Paper:**

1 (Minor). The work is mostly empirical, and there is a large number of parameters/design choices whose setup is not always easy to understand. However, I think this is an inherent limit of any neural-network learning mechanism. It is hard to find some general principle from this work.
2 (Minor). Redundancy in the description of the modules. All modules share the very same structure (the sigma function relating the input with the weights and the bias); I feel this can be factorised and space can be recovered to better explain some design choice.
3 (Minor). The work may be a little incremental w.r.t. ASNET for classical planning

---

> ### Author Rebuttal · Authors · 2024-01-28
>
> Q1: We had introduced the preconditions and goals as conjunctions in an attempt to simplify the exposition. In reality, as with the original ASNets, the preconditions can be any quantifier-free formula involving conjunction, disjunction and negation of propositions and numeric comparisons, and the relatedness definition support these without change. Note that the experimentation domains do not have preconditions that are more complex than conjunctions.
> For goals, the main requirement is for there to be a consistent structure across problem instances. The network requires this such that experiences learned from reaching one instance's goal are applicable to reaching the goal of unseen instances. This is the case for blocks grouping and all other domains used in our evaluation.  We will clarify this.
>
> Q2: We would like to emphasize that the network output is a stochastic policy, so tie-breaking is conceptually external to the network, and only a part of how the network output is applied. In our case, we order actions by using the lexicographical ordering of a custom unique string representation of the actions, and break ties by preferring the first maximum-probability action.
>
> Q3: The policy will not assign any inapplicable action a non-zero probability, thanks to the masked softmax layer. See equation around line 348: if an action a is inapplicable in the current state, the boolean m_a will be set to zero. This is also the output that is used in the loss function, and therefore used to train the network.
>
> Hyperparameters: See Reviewer AtH6/Q1

---

### Official Review · Reviewer_AtH6 · 2024-01-19

**Significance And Importance:** 2
**Soundness:** 3
**Novelty:** 2
**Clarity:** 3
**Overall Evaluation:** 1
**Confidence:** 4

**Weaknesses:**

1: Minor weaknesses that are easily fixable.

**Contributions Of The Paper:**

The starting point of this work are the Action Schema Networks (ASN), a deep learning architecture for learning action policies in the context of generalized planning.  These networks exploit the relational structure of planning tasks at lifted representation, making it possible to share learned weights, but instantiated for different problems in a domain.
The contribution of this paper is the extension of ASNs to handle numeric planning problems described in PDDL2.1. This includes adding new modules for reasoning with (1) numeric fluents and (2) comparison of numeric expression. The paper also contributes with a learning algorithm that offer better control over the exploration versus learning balance.  The evaluation in some IPC domains verifies the capability of solving numeric planning problem in a competitive way and demonstrating the knowledge transfer from small to large problems.

**Ethical Considerations:**

(1) Not Applicable: The paper does not have any ethical considerations to address

**Nomination For Best Paper:**

No

**Questions For Authors:**

The hidden representation size was set to 15 and no further discussed.
- How this size relevant regarding the amount of information the ASN can encode?
- Should we expect very different results with other d_h values?

**Reproducibility:**

4: Authors promise to release code and domains (whichever apply).

**Strengths Of The Paper:**

Most works on learning generalized policies do not handle numeric reasoning or at most some basic ways of quantification.  The main strength of this work is that it adapts an existing technique to deal with a more realistic planning framework.  In addition, the result analysis covers several key questions such as the effect of combining different versions of new modules.

The network ability to share weights between problems of different sizes is something attributable to the original ASNs. But, finding the representation to extend ASNs while keeping this key property is new, and relevant for the symbolic-to-NN knowledge encoding.

**Weaknesses Of The Paper:**

Authors mentioned they didn’t experiment with IPC-2023 domains where they cannot find a configuration for the teacher planner. On the one hand, the numeric track repository contains 20 domains, so authors could have discussed the limitation that evaluation only shows part of the picture.
On the other hand, the pre-selection of domains is related to the availability of good training instances and not from a limitation of the system itself. In general, IPC problem sets are designed to be challenging for state-of-the-art planners, so using the first instances may not be feasible if they cannot be solved for exploration.  Instead, one could create instances specifically for training purposes. For instance, selecting a sub-set of goals, discarding part of the objects, or using the random problem generator, which is available for some cases. As an example, the Counters domain was run with a different set.

Minor details:
L.672: impactically -> impractically
Line 451: at lease one state -> at least one state
Figure 3: The legend, shared by the two images, can be placed outside to avoid hiding some points behind it.

---

> ### Author Rebuttal · Authors · 2024-01-28
>
> Q1: Our hyperparameters are generally selected by performing minor tuning on top of the original ASNets hyperparameters. Specifically, we used a hidden representation size of 15, while the original ASNets had a hidden representation size of 16. These sizes are around the smallest size before the network becomes unable to learn a good policy. There did not seem to be much benefit to values above 16, but we unfortunately did not have sufficient computational resources to perform a thorough investigation.
> We also inherited the number of layers from the original ASNets. We find that increasing the number of layers is not always beneficial, especially as more layers imply a larger and more computationally expensive network. We find three action layers and two state layers to be a reasonable choice that balances between network size and capability.
>
> Creating training specific instances: for a few domains, we attempted to do this. Unfortunately, in order to match our criteria for being usable as training instances (specifically solvable by some teacher configuration quickly), we had to simplify the created instances to the point that they were too simple to be representative of the evaluation instances and useful for training.

---

### Official Review · Reviewer_AMik · 2024-01-19

**Significance And Importance:** 2
**Soundness:** 3
**Novelty:** 2
**Clarity:** 4
**Overall Evaluation:** 1
**Confidence:** 5

**Weaknesses:**

0: Minor weaknesses requiring some work to be addressed for the paper to be accepted.

**Contributions Of The Paper:**

This is the first work to learn generalized policies for the numerical fragment of PDDL 2.1 using the numerical factors as first-class citizens. Previous work either generated generalized policies from integer linear problems (not PDDL), computed them using a specific target language, e.g. Description Logics, using the numerical factors only qualitatively, or computed generalized policies as programs but factors were in the integer domain (no real numbers). The contributed generalized policies are learned in a deep learning architecture, which extends the propositional fragment of some previous work named ASNets. The design choices for the architecture have been analyzed with an ablation study and a comparison with a state-of-the-art numerical planner, and the results have been obtained with the most recent IPC23 - Numerical Track benchmarks.

**Ethical Considerations:**

(1) Not Applicable: The paper does not have any ethical considerations to address

**Nomination For Best Paper:**

No

**Questions For Authors:**

Q1. Line 153, when the policy is defined as \pi^\theta(a|s) over *all* applicable actions. Does the output size require to be at least as large as the number of ground actions?
Q2. What do you mean by different v-ASNets at the same layer l (Lines 209-210)?
Q3. Why do you say that same weights are used to any instance in a domain (Lines 211-213)? The number of weights seems to depend on the number of grounded actions, but this changes from instance to instance (Lines 190-196).
Q4. The "action count c_a helps the network avoid cycling between adjacent states" -> is this included in any kind of loss function? otherwise, why a counter prevents cycling between adjacent states?
Q5. Line 374, what do you mean by "skip connections"? how is this implemented?
Q6. "Actions leading to the lowest-cost plans are labelled optimal" -> How do you know an action belongs to a lowest-cost plan? Are you running an optimal planner?
Q7. I don't find the difference between the "stale" problem (Line 421), and "repeatedly remove the oldest group" (Line 456). The latter will also consider the more recent states instead of the old ones, so that looks like the same problem.

**Reproducibility:**

4: Authors promise to release code and domains (whichever apply).

**Strengths Of The Paper:**

* Handles more expressive input features than previous deep learning approaches for computing generalized policies
* Extends previous work on ASNets by including new sort of modules for numerical values and comparisons.
* It outperforms a state-of-the-art numerical planner when using the best configuration for each domain.

**Weaknesses Of The Paper:**

# MAJOR
  * Lines 156-158, if the network architecture is instance dependent, the weights can not be instance-agnostic (at least the number of weights changes if the architecture changes), so this point is not very clear.
  * Line 305, first time that uses landmarks without any previous definition, citation or how they were computed. Also, it is hard to know the impact of those landmarks in the resulting architecture.
  * Line 358, ties are deterministically broken, but it does not explain how this deterministic process works.
  * The article explains that focus on specialized modules like C-ASNets or F-ASNets reduces the computational burden and potentially the overfitting, however, there are no justifications. Regarding computational burden, it is not explained if it applies to training, evaluation or both. Regarding overfitting, it seems wrong to consider that only "fluent modules" should overfit less than "fluent+comparison modules", e.g. let's pose a problem that can be generally solved only with comparison modules, then only fluent modules should overfit more to the training instances than "fluent+comparison modules" which may focus, during the learning phase, on the comparison modules of the architecture to generally solve the problem.
  * Three action layers & two state layers do not look very deep, however, the depth is one of the key factors that affect the reasoning performance in previous ASNets (receptive field problem). It would be good to justify in more detail this design choice.
  * When combining both (fluent+comparison modules) the performance is not as good as expected. I cannot get why the neural network does not learn to discard the modules that do not help to generalize. In my opinion, this experiment should outperform the specialized ones. (Similar comment for the question/answer between Lines 610-620).

# MINOR
  * The article explains each architecture module and then the input and output to the network as special cases. I find this order a bit confusing, because the input to the network already appears for the first action layer, but input to action modules are explained as intermediate hidden representations. Introducing the original input before action modules would help the reader.
  * In definition of v^l_f (after line 235), there are missing ')' at the end of 'R' symbol
  * "v-ASNetsto" -> "v-ASNets to"
  * In the experimental setup, when only comparing heavily numeric & hybrid domains, I guess this is because there are no "heavily propositional" (i.e. classical) domains.
  * "For fairness, we enable action optimality approximation with the original algorithm." -> I did not understand this part.

---

> ### Author Rebuttal · Authors · 2024-01-28
>
> Q1: Yes
>
> Q2-3: ASNet learns one set of weights per domain. The network is different for each instance but weights only depend on the domain's action schema and predicates: at each layer, ASNet enforces that all propositions that are instances of the same predicate share the same weights, and all actions that are instances of the same action schema share the same weights. Note such actions have the same number of related propositions, hence their weight matrices will have the same dimensions. For propositions, we additionally need to use pooling to ensure matrices have the same dimension (224-244). We will clarify.
>
> Q4: Action counts are not included in the loss function, but only in the network input. They allow learning a non-stationary policy applying different actions depending on what actions it has applied before. Action counts were already included in the original ASNet, and are shown to be empirically beneficial in Toyer et al 20.
>
> Q5: There is a connection from each action module at layer l to the module for the same action at layer l+1. This is implemented by having the last element of $u^{l+1}_a$ be the output of $\phi^l_a$. Similarly for fluents, the last element of $v^{l+1}_f$ is $\psi^l_f$. We omitted those in the equations for simplicity.
>
> Q6: We do not always use an optimal teacher. The "optimal" action is the action chosen by the teacher. We will rephrase.
>
> Q7: The list 1…3 explains the issues we face, and the paragraphs below our solutions to these. So Line 456 describes the solution to line 421.
>
> Performance of F+C: Including only one of fluent and comparison modules reduces the computational burden as the network has fewer modules and can propagate vectors faster. In turn this potentially allows more to be learnt within a given training time. We only hypothesized that including both module types can worsen overfitting or disallow the network to focus on a particular form of numeric reasoning. In fact the worse performance of F+C came unexpected to us as well and is an interesting topic for future investigation.
>
> Landmarks: Experimentally, we find that removing landmarks from the network input may impact performance. For TPP and Delivery, the impact is not significant, whereas for Block Grouping, the network heavily relies on the numeric landmark information. In retrospect, we should have provided landmark related results in the supplementary material and will do so in the final paper.
>
> Network depth: See AtH6/Q1
> Determinism: See cD32/Q2

---

### Meta-Review · Area_Chair_JNdy · 2024-02-04

**Recommendation:** Accept (Oral)
**Confidence:** 5

**Metareview:**

The paper presents an interesting extension of the ASNet architecture for generalizing numerical planning. Please, read the reviews as some details of the architecture, and the experiments are not fully clear.

Also, the loss function being minimized is not given, only briefly mentioned in the last sentence of the first paragraph of the section "Dynamic Exploration". I believe it is the same as in the ASNet (journal) paper, but this must be given for the paper to be self-contained and reproducible.

**Ethical Considerations:**

(5) Excellent: The paper comprehensively addresses all of the applicable ethical considerations